# A Laccase Gene Reporting System That Enables Genetic Manipulations in a Brown Rot Wood Decomposer Fungus *Gloeophyllum trabeum*

Weiran Li,[a] Charles Ayers,[a] Weiping Huang,[a] Jonathan S. Schilling,[b] Daniel Cullen,[c] Jiwei Zhang[a]

[a]Department of Bioproducts and Biosystems Engineering, University of Minnesota, St. Paul, Minnesota, USA
[b]Department of Plant and Microbial Biology, University of Minnesota, St. Paul, Minnesota, USA
[c]U.S. Department of Agriculture–Forest Products Laboratory, Madison, Wisconsin, USA

**ABSTRACT** Brown rot fungi are primary decomposers of wood and litter in northern forests. Relative to other microbes, these fungi have evolved distinct mechanisms that rapidly depolymerize and metabolize cellulose and hemicellulose without digesting the more recalcitrant lignin. Its efficient degradative system has therefore attracted considerable attention for the development of sustainable biomass conversion technologies. However, there has been a significant lack of genetic tools in brown rot species by which to manipulate genes for both mechanistic studies and engineering applications. To advance brown rot genetic studies, we provided a gene-reporting system that can facilitate genetic manipulations in a model fungus *Gloeophyllum trabeum*. We first optimized a transformation procedure in *G. trabeum*, and then transformed the fungus into a constitutive laccase producer with a well-studied white rot laccases gene (from *Trametes versicolor*). With this, we built a gene reporting system based on laccase gene's expression and its rapid assay using an 2,2′-azino-bis(3-ethylbenzothiazoline-6-sulfonic acid) (ABTS) indicator dye. The laccase reporter system was validated robust enough to allow us to test the effects of donor DNA's formats, protoplast viability, and gene regulatory elements on transformation efficiencies. Going forward, we anticipate the toolset provided in this work would expedite phenotyping studies and genetic engineering of brown rot species.

**IMPORTANCE** One of the most ubiquitous types of decomposers in nature, brown rot fungi, has lacked robust genetic tools by which to manipulate genes and understand its biology. Brown rot fungi are primary decomposers in northern forests helping recycle the encased carbons in trees back to ecosystem. Relative to other microbes, these fungi employ distinctive mechanisms to disrupt and consume the lignified polysaccharides in wood. Its decay mechanism allows fast, selective carbohydrate catabolization, but without digesting lignin—a barren component that produces least energy trade back for fungal metabolisms. Thus, its efficient degradative system provides a great platform for developing sustainable biotechnologies for biomass conversions. However, progress has been hampered by the lack genetic tools facilitating mechanistic studies and engineering applications. Here, the laccase reporter system provides a genetic toolset for genetic manipulations in brown rot species, which we expect would advance relevant genetic studies for discovering and harnessing the unique fungal degradative mechanisms.

**KEYWORDS** wood decomposition, brown rot, *Gloeophyllum trabeum*, fungal genetics, reporting system, laccase

Fungi are the primary decomposers that recycle the carbon sequestered in wood. Wood-degrading fungi are taxonomically distributed in *Ascomycota* and *Basidiomycota* phyla, and they have evolved a diverse array of enzymatic and nonenzymatic mechanisms to breakdown lignified polysaccharides embedded in wood (lignocellulose) (1). By decay

Address correspondence to Jiwei Zhang, zhan3437@umn.edu.

The authors declare no conflict of interest.

phenotypes, these fungi are categorized into white rot, brown rot, and soft-rot groups that degrade lignocellulose using different strategies (2, 3). Among these, brown rot fungi often cause faster wood degradation and selectively utilize the encased carbohydrates, leaving the modified lignin residues (4–6). Brown rot fungi are common in boreal forests and on conifer wood, where they help sequester carbon in lignin residues and are considered key microbial drivers of carbon cycles (1). In addition to their ecological significance, the robust brown rot mechanisms of these fungi also hold promise for developing sustainable bio-products and biotechnologies; the ability to unlock sugars from wood without sacrificing lignin is, among the other attributes of brown rot fungi, a potentially valuable industrial mechanism.

The mechanisms of brown rot fungi remain less clear than that of their lignin-degrading counterparts, white rot fungi. During brown rot, intensive structural loss of lignocellulose first occurs via oxidation, which leads to the depolymerization of poly-saccharides and allows glycoside hydrolases to gain access (6–10). This initial attack is considered the key step to brown rot and is known to be driven by reactive oxygen species (ROS; e.g., OH). These ROS agents are produced by Fenton redox cycles that rely on low-molecular-weight brown rot metabolites (e.g., hydroquinones) and then employed to degrade lignocellulose substrate by electrophilic attacks, nonspecifically (11–13). This dependence on aggressive ROS attacking distinguishes brown rot from white rot that uses ligninolytic peroxidases and cellulases, instead, for removing lignin and degrading polysaccharides (14, 15). Although evidence has shown ROS is the key to brown rot, it is still unclear which genes and pathways are involved.

This major gap in brown rot fungal knowledge is largely due to the lack of available genetic tools. We have an increasing number of annotated genomes, but these primarily show us what pathways have been lost, not what pathways have been gained among brown rot fungi. Wood-degrading fungal genome comparisons have revealed that brown rot fungi evolved in independent *Agricomycotina* clades multiple times and that this evolu-tion involved the loss of ligninolytic genes, specifically class II peroxidases, and the loss of substantial genetic repertories for wood degradation (15–17). This also includes many car-bohydrate-active enzymes such as cellobiohydrolases (CBH I and II). These gene losses, which exceed 60% of lignocellulolytic "usual suspects" (15), are not matched by the genetic information about what was gained to deploy ROS pathways.

To address this gap, we recently used multiomic tools to complement our understand-ing of brown rot genetics and target genes of interest. First, we found that brown rot fungi can stagger gene expression to first produce ROS for initial decay, and then express glyco-side hydrolases for saccharification (10, 18). This, combined with cross-species comparisons, has since enabled us to discover the decay-stage-dependent genes and shuffled genetic regulatory system unique to brown rot (19, 20). Supporting this, we also discovered that brown rot adopts an attenuated glucose-mediated repression system to boost cellulase production for rapid carbohydrate release (21). These studies have pinpointed potential genetic inventories adapted by brown rot for rapid wood decomposition but have been awaiting further exploration by genetic manipulations that can validate function.

With this context, here, we shared a genetic transformation system that we devel-oped in a model brown rot fungus, *Gloeophyllum trabeum*, which has been widely used for over 50 years to study the mechanisms of brown rot (5, 12). Following success in transformation, we then built a gene-reporting system that relies on laccase as the reporter for further genetic platform optimization. This work addresses an increasing interest in decoding the genetics in wood decay fungi, especially for brown rot fungi in which no genetic platform has been made available, to date.

## RESULTS

**Genetic transformation of the brown rot fungus *G. trabeum*.** We successfully transformed the model brown rot fungus *G. trabeum* by delivering a hygromycin-resistant gene *Hyg^R* into protoplast cells (Fig. 1). To prepare protoplasts, we blended two commercial enzymes (i.e., Yatalase and VinoTase Pro) that contain the necessary enzyme components for

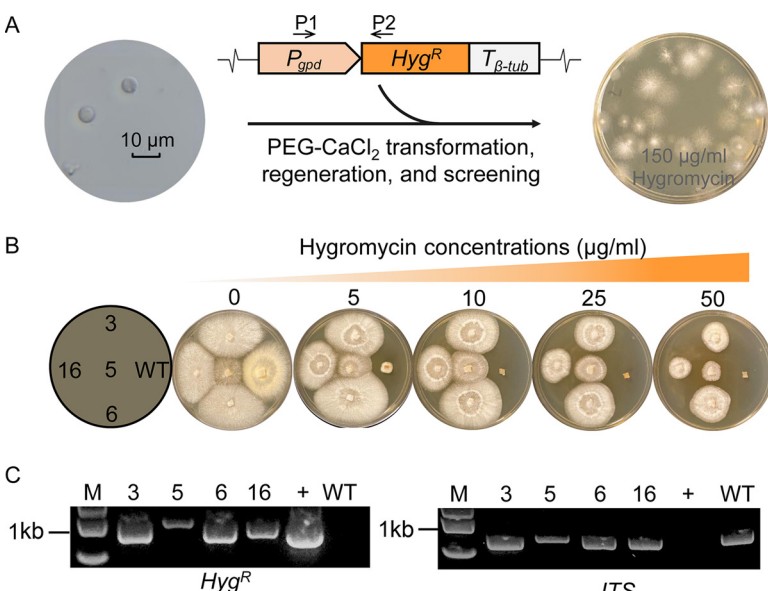

**FIG 1** Genetic transformation of the brown rot fungus *G. trabeum* Mad. 617R. (A) Protoplasts obtained from *G. trabeum* mycelia were transformed with a hygromycin-resistant gene (*Hyg^R^*) via a modified PEG-CaCl₂-mediated transformation method. P1 and P2 indicate the loci of gene-specific primers for PCR verification of transformants. (B) Transformants were confirmed for their resistance to low and high concentrations of hygromycin, while the wild-type (WT) strain maintained the high susceptibility on the same plates. Diagram on the far left indicates the loci of the inoculated strains. (C) Transformants were further validated via PCR by amplifying the *Hyg^R^* gene to confirm its genetic insertion and by amplifying the *ITS1* gene to double check PCR conditions. +, plasmid containing the *Hyg^R^* cassette. 3, 5, 6, and 16 are transformants.

fungal cell wall digestion, which enabled us to obtain 2~10 × 10⁷ protoplasts from 10 g fresh (hydrated) hyphal biomass, an adequate level for the downstream transformation. Viability testing showed that 1 to 2% protoplasts could survive and regenerate on either Highley's minimal medium (HMM) or YMG medium (Yeast extract 4 g/L, Malt extract 10 g/L, Glucose 10 g/L) containing 0.5 M sucrose as the osmotic stabilizer (see Fig. S1 in the supplemental material). Regenerated colonies (*n* = 100) showed the same phenotype during the successive plate cultures, suggesting that the protoplasts produced with the current procedure were genetically stable homokaryons.

The *Hyg^R^* gene driven by a *Rhodonia placenta* glyceraldehyde-3-phosphate dehydrogenase promoter (*P_gpd_*) and a β-tubulin terminator (*T_β-tub_*) was constructed and applied for transforming *G. trabeum* protoplasts (Fig. 1A; Fig. S2A). The cross-species promoter and terminator were chosen to control *Hyg^R^* expression according to previous work (22). To screen the transformed protoplasts, an overlaying medium containing 150 μg/mL hygromycin was optimized to produce least aborted transformants (Fig. 1A; Fig. S2B). Five rounds of successive culturing showed that the selected transformants were significantly resistant to up to 50 μg/mL hygromycin, while the wild-type strain was fully inhibited at 10 μg/mL and above, indicating that the transformants were mitotically stable (Fig. 1B). Successful insertion of the *Hyg^R^* gene into the genome was confirmed by PCR of four randomly selected transformants (Fig. 1C). Statistically, using this *Hyg^R^* gene for transformation could allow us to obtain 14 to 18 transformants per 10⁷ protoplasts per μg DNA (Fig. S2C).

**Heterologous expression of a white rot laccase in the brown rot fungus, *G. trabeum*.** With the established transformation procedure, we then expressed in *G. trabeum* a laccase gene *TveLac3* (Trave1|138531; NCBI accession: XM_008034423) originating from the white rot fungus *Trametes versicolor*. Laccase (EC 1.10.3.2) is a multicopper-containing oxidase catalyzing the one electron oxidation of substrate by O₂, and there has been a reliable, rapid colorimetric method to detect this enzyme via oxidizing the colorless ABTS to a green end product (23, 24). Given this easy-to-detect feature, we intended to express a heterologous laccase gene for a rapid test of our transformation procedure.

*P_gpd_* promoter and *T_β-tub_* terminator of *R. placenta* were applied to drive the expression

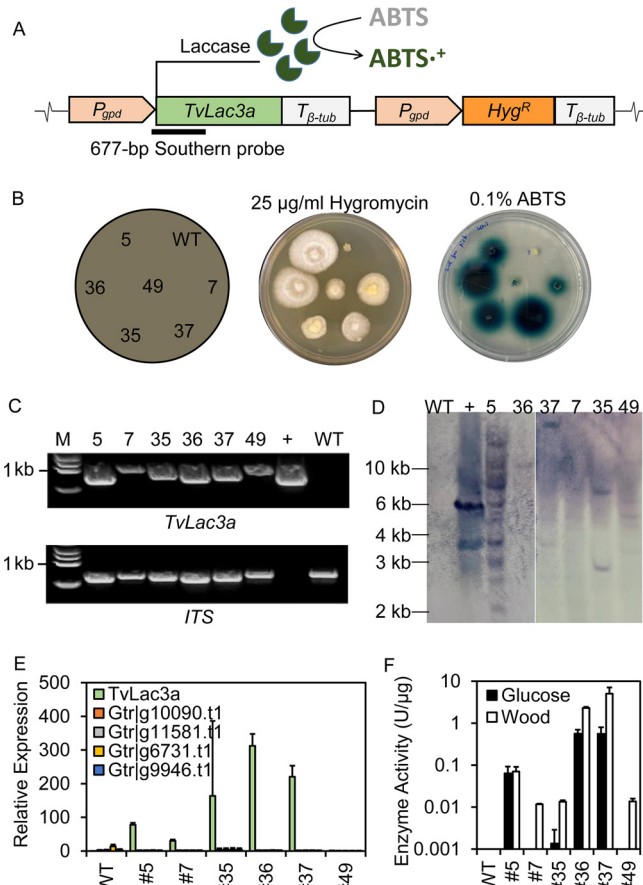

**FIG 2** Heterologous expression of a white rot laccase gene in *G. trabeum*. (A) Full construct for expressing the laccase-encoding gene *TveLac3* (from white rot fungus *Trametes versicolor*) and the diagram for detecting laccase production by oxidizing ABTS. Bold line indicates the 677-bp probe fragment used for Southern blotting of laccase transformants. (B) Double-phenotypic confirmation of six randomly selected laccase transformants on 25 μg/mL hygromycin and 0.1% ABTS plates. Diagram on the far left indicates the loci of the inoculated strains. Successful integration of laccase expression cassette into transformant genome was further confirmed by PCR of *TveLac3* and *ITS* genes (C) and by Southern blotting (D) that showed various copy numbers of *TveLac3* gene (1 to 9) have been inserted at different genomic loci. (E) RT-qPCR showed high mRNA levels of the heterologous *TveLac3* gene after 4 days of growth in HMM-glucose media, while the 4 endogenous laccase genes kept very low expression levels. (F) Enzyme assays showed that transformants, relative to wild type, had significant ($P < 0.05$, paired $t$ test) laccase production in either glucose or wood media. Laccase activity in crude enzyme was normalized to the total protein amount and was expressed as U/μg total proteins. +, plasmid containing the *TveLac3* and *HygR* cassettes; WT, wild-type *G. trabeum*; 5, 7, 35, 36, 37, and 49 are laccase transformants. Mean ± SD values from three bioreplicates are presented.

of *TveLac3* because of their successful use in expressing the *HygR* gene (Fig. 2A). The vector containing both *TveLac3* and *HygR* genes was delivered into *G. trabeum* protoplast, and transformants were obtained by screening their resistance to hygromycin B (Fig. 2B). By the following ABTS plate assay, laccase production was then quickly confirmed in the transformants, but not in wild-type *G. trabeum* (Fig. 2B). Genomic insertion of the *TveLac3* gene was confirmed by both PCR and Southern blotting, with 1 to 9 copies of *TveLac3* at different insertion loci detected in six transformants (Fig. 2C and D). These results have thus validated our success in transforming *G. trabeum* with the *TveLac3* gene.

To further detect the expression of *TveLac3*, we submerged cultured transformants in HMM medium. RT-qPCR results showed that *TveLac3* mRNAs were only detectable in transformants, but not in wild-type, and its levels were somehow variable among six selected transformants when grown on glucose (Fig. 2E). Notably, the mRNAs of four native laccase-like genes were rarely detected in both wild type and transformants, which confirmed that none were influenced by the transgenic expression of *TveLac3* (Fig. 2E). Consistent with

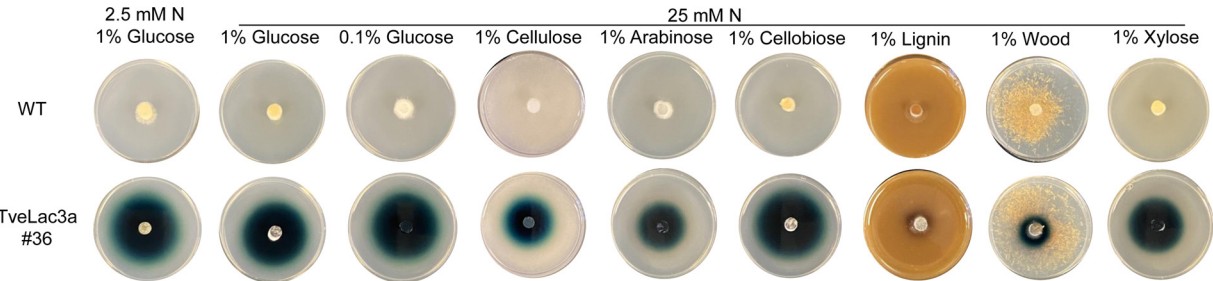

**FIG 3** Laccase production of *G. trabeum* and its *TveLac3a* transformant under varied carbon and nitrogen conditions. To induce laccase production, the wild-type strain (WT) and a transformant TveLac3a no. 36 were incubated onto HMM agar plates supplemented with varied C and N nutrients for 10 days at 28°C. Mycelium was then transferred to a 0.1% ABTS plate (pH 4.5) to measure laccase activity by recording the color changes after 3 days of incubation at 28°C. One representative of three replicate experiments was reported here.

the *TveLac3* mRNA levels, laccase activities were detected extracellularly for transformants grown on glucose, but not in the wild type (Fig. 2F, Fig. S3A; $R^2 = 0.78$). Slightly different than on glucose, all transformants showed higher laccase activities on wood substrate, perhaps due to the different culture conditions.

To assess if ABTS plate assay could be a rapid method for quantifying laccase expression in transformants, correlation analysis was conducted between ABTS halo size and mRNA levels. Our analyses showed that the halo size linearly reflected the *TveLac3* mRNA levels ($R^2 = 0.90$ by linear regression; Fig. S3B) and therefore could be used as an indicator to rapidly evaluate laccase expression in transformants.

Together, we successfully converted *G. trabeum* into a constitutive laccase producer by transforming it with a white rot laccase gene *TveLac3*. The production of this heterologous laccase was confirmed by evidence ranging from gene insertion and transcription to enzyme secretion in both solid and liquid cultures. Importantly, given the rapid laccase detection method, *TveLac3* showed the promise as a reporter gene for further genetic platform optimization in *G. trabeum*.

**The native laccase-like genes of *G. trabeum* are silenced, not interfering with the laccase-reporting system.** Although four native laccase-like genes were revealed in the wild-type *G. trabeum* (15), none of them were validated and showed the enzyme functions, as in other soft rot and brown rot fungi (25). RT-PCR results validated that all four genes were rarely expressed in the wild-type strain (Fig. 2E), which was further confirmed by its nondetectable laccase activities on varied nitrogen and carbon source conditions (Fig. 3). Differentially but as expected, the laccase activities were constitutively expressed in the *TveLac3a* transformant. Thus, we confirmed that the native laccase genes of *G. trabeum* have been silenced and won't cause interference on the application of the laccase-reporting system.

**Genetic platform optimization with the laccase-reporting system.** Combining *TveLac3* with the ABTS plate assay as a system to report gene expression, we investigated several key factors controlling genetic manipulation in *G. trabeum*. Three *TveLac3* constructs (RE1, RE2, and RE3) controlled by different promoter/terminator combos were built and applied for these tests, with the promoter and terminator regions retrieved from three highly expressed housekeeping genes *Rplgpd*, *Rplβ-tublin*, and *Gtrgpd* (GSE108189) (Fig. 4A) (26). After transforming these constructs into *G. trabeum*, 28, 104, and 12 hygromycin-resistant transformants were obtained for RE1, RE2 and RE3, respectively. Among these, only 75 to 95% tested PCR positive by amplifying the *Hyg^R* gene. Surprisingly, almost all the PCR-positive transformants were ABTS positive, suggesting that the laccase reporter was more robust for transformant screening (Fig. 4B).

As expected, the ABTS halo measurements (size $\approx$ hundreds of mm²) suggested that all the three promoters tested here were strong, constitutive regulatory elements. Laccase activities found in each vector group were more variable, perhaps due to the random insertion or multiple copies of the *TveLac3*. Despite this, statistics showed that *Rplβ-tublin* was the strongest promoter and was significantly different than that of *Rplgpd* (ANOVA test, $P < 0.05$) (Fig. 4C).

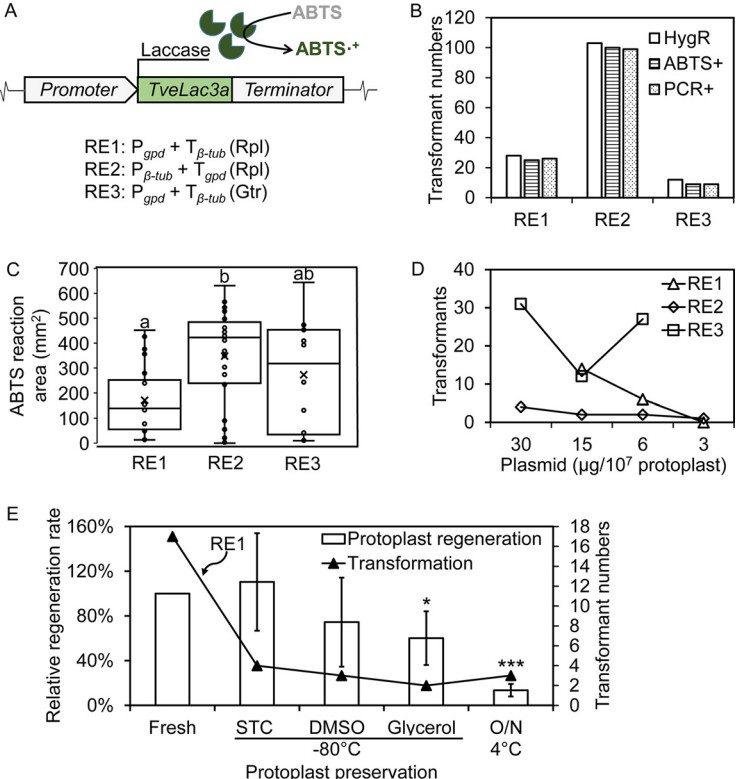

**FIG 4** Laccase-reporting system for optimizing the genetic platform in *G. trabeum*. (A) The laccase-reporting system that was composed of a *TveLac3* gene and was driven by variable regulatory elements (RE). Three constructs (RE1, RE2, and RE3) built with endogenous (Gtr, *G. trabeum*) or cross-species (Rpl, *Rhodonia placenta*) promoters/terminators were used to test the performance of the reporting system. (B) Validation of RE transformants by three different methods (hygromycin-resistance, ABTS oxidation, and PCR of *TveLac3*) indicated that the ABTS plate assay was consistent with the PCR test and could provide faster, more reliable screenings on transformants. (C) The strength of three promoters in RE constructs in driving laccase expression was estimated by measuring the ABTS halo area's size on plate. More than a dozen transformants were used for each construct (RE1 = 21, RE2 = 31, RE3 = 12) for a more accurate evaluation. Different letters indicate significant differences ($P < 0.05$, ANOVA test) between RE constructs. (D) Transformation efficiencies with different donor DNA amounts were estimated. Three RE constructs were used to represent independent replicates. (E) Influences of protoplast preserving methods on transformation efficiency were tested using construct RE1, which indicated that preservation treatment could influence transformation through affecting either protoplast regeneration rate or competence. Paired *t* tests were used for significance calculation (*, $P < 0.05$; ***, $P < 0.001$). Protoplasts were preserved for 1 week in STC buffer, DMSO, or glycerol at −80°C or overnight (O/N) in STC at 4°C. Mean ± SD values from three bioreplicates are presented. The regeneration rates of fresh protoplasts (1.01 ± 0.47%) were set as 100% for normalization.

The effects of donor DNA amounts (3, 6, 15, and 30 µg) on transformation efficiency were investigated with the above three constructs (Fig. 4D). In general, the results indicated that at least 6 µg DNA would be required to ensure a successful transformation and that increasing DNA amount could result in more transformants. Interestingly, these results also indicated that different constructs could lead to large variability in transformation rates, with RE3 generating the lowest number of transformants. This difference is even more significant compared with transforming the *Hyg*[R] gene solely (Fig. S2), indicating that transformation rates might also be influenced by donor DNA size or higher-order structure in addition to by DNA amount, as in other microbes (27).

With the vector RE1, the effects of protoplast preservation on transformation rate were also investigated. Four different preservation treatments, including STC buffer, DMSO, or glycerol at −80°C for 1 week and STC buffer at 4°C for overnight, were compared with fresh protoplast for their transformation efficiencies. The results showed that transformation efficiencies were significantly decreased in the preserved protoplasts, and this may be because the preserving process can lower the regeneration rates (glycerol at −80°C and STC at 4°C) or interfere with protoplast cell competence

(Fig. 4E). Nevertheless, the preservation step might still be helpful for boosting genetic manipulation throughput, albeit with a sacrificed transformation rate.

## DISCUSSION

In this research, we have overcome a major obstacle in understanding a globally important decomposition pathway, brown rot, by establishing the laccase reporter-based genetic manipulation system for the fungus *Gloeophyllum trabeum*. Brown rot fungi play a key role in boreal forest carbon cycles by returning photosynthetically sequestered carbon to the atmosphere (as carbon dioxide) and to soils (as humic substances and dead fungal biomass) (1). Their fast, carbohydrate-selective decay mode could "steer" carbon flow routes in forests by shunting most of the nondigestible lignin residues into the inactive soil carbon reservoir rather than into $CO_2$. Evidence suggests that brown rot fungi adopted unique machinery to bypass the lignin barrier to access sugars and that this fungal adaptation has been largely due to genetic reshuffling. During brown rot evolution, lignocellulose-degrading genes and gene regulation systems have both been greatly altered, relative to white rot-type ancestors (10, 15, 20, 21). However, interrogation of this mechanism and its evolution have been severely limited due to the lack of genetic tools that allow gene manipulations in brown rot species. Compared with the previous trials on transforming brown rot species for molecular breeding for lignin degradation (28), here, we provided a gene reporting system along with more complete details on how to establish a robust genetic manipulation in *G. trabeum*. Specifically, we provided an optimized transformation procedure and the genetic regulatory elements such as promoters and terminators that can be used for driving gene expression and learning how brown rot fungi mineralize wood.

Genetic manipulation has already been made available in some wood-decomposing white rot fungi (e.g., *Phanerochaete chrysosporium* and *Pleurotus ostreatus*) (29–31), but the similar success was rare in brown rot species due to the uniqueness of these fungi. The lack of spores in the laboratory conditions makes it difficult to attain active brown rot fungal protoplasts for transformation using the freshly germinated cells, a standard procedure for *Phanerochaete chrysosporium* and many Ascomycete fungi (31, 32). Instead, we succeeded in obtaining enough viable protoplasts ($10^7$ cells with the 1 to 2% regeneration frequency) by using the actively growing vegetative mycelia of *G. trabeum* for protoplasting for a successful transformation of a marker gene $Hyg^R$ (Fig. 1). Genetically unstable, heterokaryon protoplast cells may be released by the digestion of vegetative mycelia (33), but we avoided this by using the *G. trabeum* monokaryotic strain ATCC 11539 (Mad. 617R) as the host. The phenotypic checking of hundreds of regenerated colonies confirmed that the protoplasts were all homokaryon (data not shown).

The following expression of the white rot laccase in *G. trabeum* not only validated our transformation procedure, but also allowed us to build a gene-reporting system. Reporter genes such as green fluorescent protein (GFP), $\beta$-galactosidase (LacZ), and $\beta$-glucuronidase (GUS) have been developed in many fungal species for genetic platform optimization (34–36) but grafting them into a brown rot fungus could be prohibitively challenging. First, either the endogenous $\beta$-galactosidase/$\beta$-glucuronidase genes or the incompatible expression system of brown rot fungi make it difficult to express these reporters. Second, the methods for detecting these reporters are often time consuming and would need expensive fluorescence microscopy to validate. These disadvantages could be circumvented by using laccase as the reporter in brown rot fungi; i) Expressing the basidiomycete laccase genes is readily compatible with the brown rot fungal system, and ii) the laccase detection that relies on a visible, colorimetric ABTS-oxidizing method is relatively inexpensive and fast (25, 37). Moreover, by measuring laccase expression in the wild-type *G. trabeum* strain, we excluded the potential interference of native laccase-like multicopper oxidase genes (38) (Fig. 2E and F). Thus, we validated that the white rot laccase gene can be a reliable reporter for brown rot genetic manipulations. Indeed, our data showed that the use of laccase reporter for screening transformants was faster and more reliable relative to hygromycin selection.

Using the laccase-reporting system, we tested the effects of several factors on transformation efficiency in our *G. trabeum* system. We found both the quantity and quality of

protoplasts will influence *G. trabeum* transformation, as in other fungal systems (22, 29). In our study, using less than $1 \times 10^7$ protoplasts in 1 mL transformation system dramatically decreased transformation rates. Cryopreservation could not keep the high transformation rates of the fresh protoplasts due to the decrease in either the protoplast viability or competence (Fig. 4E). Overall, this aligns with similar research in *Coprinopsis cinerea* (39), in which cryopreservation processes would cumber both survival and transformation of the protoplasts. Success in *G. trabeum* transformation was still obtained using the preserved protoplasts, but at lower rates.

Donor DNA amount and size can also influence the transformation in *G trabeum*. Here, we found more transformants were obtained as the DNA amount increased, and this trend did not plateau until $\geq$30 $\mu$g DNA was applied (Fig. 4D). We also found larger DNA size could dramatically decrease the transformation efficiency in *G. trabeum*. The number of transformants per $\mu$g DNA per $10^7$ protoplasts was decreased from 13.7 to 17.9 for pGHT vector (contains only a *HygR* cassette; ~5 kb) to 0.26 to 1 for laccase vector (contains both *HygR* and *Lac3* cassettes; ~10 kb). This level matches with that of *Flammulina velutipes* Fv-1 and *Dichomitus squalens* CBS464.89, in which 0.1 to 0.3 and 0.25 to 2.6 transformants were obtained, respectively, with the 7-kb and 9-kb plasmids (22, 40) (Table S3). When an even larger sized plasmid (14 to 17 kb) was used in *Cerrena unicolor* BBP6, the transformant number dropped to only ~0.1 (41). Regarding this, we recommend applying more DNA for high-throughput transformation to compensate the effects of larger size, when one single vector must be built for simultaneous manipulation of multiple genes.

The laccase-reporting system also provides a tool to discover promoter function. Here, we validated three strong, constitutive promoters obtained from housekeeping genes *Rplgpd*, *Rpl$\beta$-tublin*, and *Gtrgpd* for their functions in driving heterologous gene expression in *G. trabeum* (Fig. 4C). Similar to the previous research in white rot fungi (22, 42), it is not surprising that in our work these three promoters showed strong and similar strength. However, it has been challenging to accurately quantify their strength due to the variable expression levels among transformants. We learned, in addition to promoter function, both the copy numbers and insertion loci of the *TvLac3* gene in transformants can influence its expression levels, as in our own results (Fig. 2) and in other fungi (43). To eliminate the interfering effects on gene expression, here we opted to measure the averaged strength of a large number of transformants for the comparison. Although not ideal, this strategy allowed us to calculate the significant difference among three promoters. An even more effective approach would be using homologous recombination (HR) for a laccase gene expression with defined insertion locus and copy number. This inevitably will need more understanding of HR and nonhomologous end-joining (NHEJ) mechanisms and gene-editing tools development in *G. trabeum* and other brown rot *Basidiomycetes*, although there are still many challenges that hinder genomic manipulations in these species (summarized in review [30]). To overcome these difficulties, we are attempting to develop a CRISPR-Cas9-based genome editing system in *G. trabeum*. The laccase-reporting system described here provides a foundation for the development of genome-editing tools, by which we anticipate discovering and harnessing the unexplored wood-decomposing mechanisms employed by brown rot fungi.

## MATERIALS AND METHODS

**Fungal strains and culturing conditions.** *Gloeophyllum trabeum* (Pers.: Fr.) Kars. Mad. 617R (ATCC 11539) was obtained from the USDA Forest Products Laboratory (Madison, WI, USA) and was used for genetic manipulation in this study. This strain represents the *Gloeophyllales* clade of Basidiomycetes. The annotated genome for this strain (https://mycocosm.jgi.doe.gov/Glotr1_1/Glotr1_1.home.html) is publicly available and suggests that it is a monokaryotic strain. *Rhodonia placenta* Mad. 698R-SB12 (https://mycocosm.jgi.doe.gov/PosplRSB12_1/PosplRSB12_1.home.html) was the source for promoters and terminators used for driving cross-species gene expression in *G. trabeum*. *Trametes versicolor* A1-ATF, a white rot fungus, was used for cloning the laccase gene from cDNA using the JGI's genome (https://mycocosm.jgi.doe.gov/Trave1/Trave1.home.html).

The YMG (Yeast extract 4 g/L, Malt extract 10 g/L, Glucose 10 g/L) agar was used to maintain and grow *G. trabeum* at 28°C. Active mycelia for protoplast cell preparation were obtained by inoculating a subdivided (subsectioned) plate into 100 mL YMG liquid medium and culturing for 4 to 5 days at 115 rpm and 28°C with a 12h/12h photoperiodic cycle. A YMG plate containing hygromycin B (Roche

Diagnostics GmbH, Mannheim, Germany) was used for screening *G. trabeum* transformants. YMG/HMM plates containing ABTS (2,2′-Azino-bis[3-ethylbenzothiazoline-6-sulfonic acid] diammonium salt) (Sigma-Aldrich, St. Louis, MO, USA) were used for monitoring laccase production in transformants. Highley's minimal medium (HMM) supplemented with required carbon sources was used for monitoring laccase production of the transformants in both liquid and solid cultures (44).

**Gene expression cassettes.** To generate the fungal transformation vector containing a hygromycin-selective marker gene, plasmid pGHT was constructed using the pUC19 backbone. The sequences of the glyceraldehyde-3-phosphate dehydrogenase gene's (GAPDH; JGI ID: PosplRSB12_1|1041472) promoter and the $\beta$-tubulin gene's (JGI ID: PosplRSB12_1|1181067) terminator were obtained from *Rhodonia placenta* Mad. 698R-SB12 (29). Specifically, the 1,100-bp GAPDH promoter and the 800-bp $\beta$-tubulin terminator were amplified from *R. placenta* genomic DNA by primer set GAPDH-HY-F1/R1 and GAPDH-HY-F3/R3, respectively, using Phusion high-fidelity PCR master mix (Thermo Scientific). The CDS of the hygromycin-resistant gene was amplified from pMD-1 by primer set GAPDH-HY-F2/R2 (45). PCR fragments were purified with NucleoSpin Gel and a PCR clean-up kit (TaKaRa, CA, USA) and then fused by overlap PCR to generate the Hyg$^R$ expression cassette. The overlap PCR product was then ligated into pUC19 plasmid digested with SbfI (NEB, MA, USA), using a DNA ligation kit (TaKaRa, CA, USA), to generate pGHT.

Similarly, three plasmids, RE1, RE2, and RE3, were constructed to investigate the feasibility of using different combinations of promoter and terminator, in order to drive the expression of laccase. For RE1, the *R. placenta* GAPDH promoter and $\beta$-tubulin terminator were amplified by primer sets pGHT-OV-Lac3-F1/R1 and pGHT-OV-Lac3-F3/R3, respectively. The full CDS of *TvLac3* encoding the laccase was amplified from *T. versicolor* cDNA by primer set pGHT-OV-Lac3-F2/R2. These three PCR fragments were fused by overlap PCR to generate the *TvLac3* expression cassette 1, which was then cloned into the KpnI site of pGHT to generate plasmid RE1. Similarly, the *R. placenta* $\beta$-tubulin promoter and GAPDH terminator were amplified to drive laccase expression to generate the TvLac3 cassette 2, and *G. trabeum* GAPDH gene's promoter (JGI ID: Glotr1_1|60667) and $\beta$-tubulin terminator (JGI ID: Glotr1_1|79348) were amplified to generate the TvLac3 cassette 3. The primers used for these two sets of overlap PCR were listed in the "Lac3-OV" group and the "gLac3-OV" group in Table S1, respectively. The TvLac3 cassette 2 and 3 were then cloned into the KpnI/XbaI sites on pGHT to generate plasmids RE2 and RE3, respectively.

Plasmid ligation products were transformed into *Escherichia coli* strain DH5$\alpha$ (NEB, MA, USA) and the plasmids were recovered by the PureYield plasmid miniprep system (Promega, WI, USA). All the plasmids were sequenced with Wideseq (Purdue, IN, USA) before being used for fungal transformation.

**Fungal transformation and transformants screening.** *G. trabeum* protoplast cells were made from the actively growing mycelia and then used for genetic transformation with the modified PEG/CaCl$_2$-mediated method, as previously described (22). Several key factors were first optimized to improve both the quantity and viability of protoplast cells. Specifically, the mycelia were grown in YMG liquid media less than 3 days before they were homogenized and transferred to fresh YMG for another 24 h to get the actively growing hyphal cells. Mycelia were harvested with filtration through two layers of Miracloth. Approximately 10 g wet mycelia were digested in 30 mL fungal cell wall-degrading enzymes (5 mg/mL Yatalase [Clontech Laboratories, MountainView, CA, USA] and 25 mg/mL VinoTaste [Novozyme] in 0.55 M sucrose solution) for 6 h at 30℃ and 75 rpm. Protoplasts were collected by filtration and a following centrifugation at 4℃ and 4,000 *g* for 10 min, washed twice with the ice-chilled STC buffer (1 M Sorbitol, 50 mM Tris-HCl, 50 mM CaCl$_2$, 0.5 M Sucrose, pH 7.5), and resuspended in STC to the final concentration of 10$^7$/mL. In this work, the enzymes, biomass amount, and digestion time were optimized to obtain the production of $2 \times 10^7$ to $10^8$ protoplasts from 10 g wet biomass. Protoplast viability was tested by plating $2 \times 10^6$ protoplasts mixed with the HMMG or YMG medium containing 0.5 M sucrose as the osmotic stabilizer.

Protoplasts were transformed with the PEG/CaCl$_2$-mediated method (29). Each mL of $1 \times 10^7$ protoplasts was mixed gently with 100 $\mu$L donor DNA (30 to 300 ng/$\mu$L) and 500 $\mu$L PTC (40% [wt/vol] PEG4000, 50 mM Tris-HCl, 50 mM CaCl$_2$, 0.5 M Sucrose, pH 7.5), and was incubated on ice for 30 min. Then, 6 mL PTC was added, and the mixture was incubated at 28℃ for another 30 min. YMG plates containing 0.5 M sucrose were used to regenerate protoplasts overnight and were overlaid with 150 $\mu$g/mL hygromycin B for transformant screening. Dozens to hundreds of transformants were visible after 7 to 10 days of incubation at 28℃, and the ones with the diameter >5 mm were isolated for further analyses. Mitotic stability of transformants was verified by at least five successive transfers on hygromycin B plates and was validated for transformant purity on nonselective plates. A few that form discrete, irregular colonies were abandoned for the downstream analyses. Genomic integration of target genes was confirmed by PCR of transformant DNA.

Several key factors influencing transformation efficiency were investigated, including the screening concentration of hygromycin B, donor DNA amount, and protoplast cryopreservation. Protoplasts were preserved at −80℃ in STC and STC supplemented with 12% DMSO or glycerol for 1 week using a Mr. Frosty cryofreezing container before viability test and transformation. By cryopreservation, we expected to increase the throughput of transformation. For each experiment, three independent bioreplicates were included and used for statistical analysis.

**Nucleic acid purification, Southern blotting, and RT-qPCR.** For PCR verification of transformants, a fast genomic DNA extraction from the *G. trabeum* plate was performed with the MightyPrep reagent (Clontech Laboratories, MountainView, CA, USA) according to the manufacture's instruction. High-quality chromosomal DNA for Southern blotting was isolated from mycelia grown in YMG liquid and was purified by liquid N2 grinding in tandem with the SDS-isopropanol-ammonium acetate extraction (46). Total RNA was extracted from liquid cultures with TRIzol (Life Technologies, Carlsbad, CA, USA) and was then purified with an Qiagen Mini RNeasy kit and on-column DNA digestion (Qiagen, Germantown, MD, USA) (21). RNA

quality was controlled by running the gel electrophoresis. Both DNA and RNA were quantified with the Qubit 4 fluorometer (Thermo Fisher Scientific Inc., Waltham, MA, USA).

For Southern blotting, 5 to 10 $\mu$g of genomic DNA from wild-type and transformant strains were digested with EcoRI and then separated in 0.8% agarose gel electrophoresis. The DNA probe was amplified from plasmid RE1 by primer set LacSouthernF/R (see Table S1 in the supplemental material). The digoxigenin labeling of the probe was performed using the DIG-High Prime DNA labeling and detection starter kit I (Roche Diagnostics GmbH, Mannheim, Germany) according to the kit manual. DNA blotting and visualization were carried out using the DIG-High Prime DNA labeling and detection starter kit I.

For reverse transcription-quantitative PCR (RT-qPCR), cDNA was synthesized from 1 $\mu$g total RNA with the PrimeScript RT reagent kit with gDNA eraser (Clontech Laboratories, MountainView, CA, USA) according to standard protocols and was then diluted by 20 times as the template. qPCR was conducted in an Applied Biosystems7900HT system (Thermo Fisher Scientific, Inc., Waltham, MA, USA) with iTaq Universal SYBR green supermix (Bio-Rad, Hercules, CA, USA). The PCR mixture contained 7.5 $\mu$L of SYBR green supermix, 1 $\mu$L of each primer (10 $\mu$M), 5 $\mu$L of diluted cDNA, and 1.5 $\mu$L of nuclease-free water. Primers with an amplification efficiency of 90 to 100% are listed in Table S1. The qPCR procedures and two internal reference genes (Prefoldin subunit 6, JGI ID: 125075; eIF3-p35 subunit, JGI ID: 122605) selected in our previous work were applied here (26). Cyclethreshold (Ct) values were calculated by Applied Biosystems SDS 2.4.1 software by setting the threshold as 0.3 and then used for transcription level estimation of laccase genes by the comparative Ct method (47, 48). Gene expression level was expressed as the fold change relative to *TvLac3* gene in transformant no. 49, set as 1.

**Laccase activity assays.** Laccase was used as an expression reporter for building the brown rot genetic platform in this work, given its fast, well-established enzyme testing method and its absence of enzyme activities in wild-type brown rot fungi. Both plate and liquid cultures were used for detecting laccase activities. Plate assays were performed on HMMG plates containing 0.1% ABTS to quickly screen the transformants that can express laccase and react with ABTS to form green halos. The size of the halo that represents laccase productivity was recorded at 12 h after inoculating strains and were measured with ImageJ V1.8.0 (https://imagej.nih.gov/ij/index.html).

For laccase production in liquid cultures, *G. trabeum* strains were cultured for 7 days in HMM with either 1% glucose or spruce wood sawdust. Crude enzymes were extracted by centrifuging the supernatants of the cultures at 4℃ and 10,000 $g$ for 10 min. Laccase activity was measured as the change of optical density at 420 nm ($OD_{420}$) in reactions of 40 $\mu$L of crude enzyme extract with 160 $\mu$L of 5 mM ABTS in 40 mM pH 3.0 sodium citrate buffer (10, 23). Total protein amount was measured by Quick Start Bradford protein assay (Bio-Rad, Hercules, CA, USA). Laccase activities were normalized to the amount of total secreted protein.

**Statistical analysis.** Two-tail $t$ tests were used for significance analyses of the relative expression and enzyme activity of laccase in transformants and the wild-type strain. One-way ANOVA and multiple comparison were used for analyses of ABTS reaction areas among RE1, RE2, and RE3 transformants. All analyses were performed with the SigmaPlot 14.5 software.

**Data availability.** The RNA-seq data measuring the gene expression profiles of four wood decay fungal species were used to guide the selection of highly expressed housekeeping genes for retrieving promoter and terminator. The RNA-seq data were uploaded to the Gene Expression Omnibus database at NCBI with the accession number of GSE108189.

## SUPPLEMENTAL MATERIAL

Supplemental material is available online only.

**SUPPLEMENTAL FILE 1**, PDF file, 0.7 MB.

## ACKNOWLEDGMENTS

This work was supported by the U.S. Department of Energy Office of Science Grants DE-SC0022151 from the Office of Biological and Ecological Research (BER) (J.S.S., D.C., and J.Z.). J.Z. was also supported by the startup funds at University of Minnesota.

J.Z. designed research. J.Z., W.L., and W. H. performed the transformation. W.L. extracted fungal RNA for RT-qPCR and performed biochemistry analyses. C.A. performed the cryopreservation experiments. J.Z. and W.L. organized the data, made figures and tables, and wrote the paper. J.S.S. and D.C. revised, commented on, and edited the manuscript writing. All authors made comments on the manuscript and approved the final version.

We declare no known competing financial interests or personal relationships that could have appeared to influence the work reported in this paper.

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
