## [Reviewer comments · Microbiology Spectrum]

Microbiology Spectrum

A Laccase Gene Reporting System that Enables Genetic Manipulations in a Brown-Rot Wood Decomposer Fungus *Gloeophyllum trabeum*

Weiran Li, Charles Ayers, Weiping Huang, Jonathan Schilling, Daniel Cullen, and Jiwei Zhang

Corresponding Author(s): Jiwei Zhang, University of Minnesota Twin Cities

Review Timeline:

Submission Date:	October 21, 2022
Editorial Decision:	November 23, 2022
Revision Received:	December 7, 2022
Accepted:	December 22, 2022

Editor: Chengshu Wang

Reviewer(s): The reviewers have opted to remain anonymous.

Transaction Report:

DOI: <https://doi.org/10.1128/spectrum.04246-22>

November 23, 2022

Dr. Jiwei Zhang
University of Minnesota
Bioproducts and Biosystems Engineering
2004 folwell ave
Saint Paul, MN 55108

Re: Spectrum04246-22 (A Gene Reporting System that Enables Genetic Manipulations in a Brown-Rot Wood Decomposer Fungus)

Dear Dr. Jiwei Zhang:

Link Not Available

Sincerely,

Chengshu Wang

Journals Department
Reviewer comments:

Reviewer #1 (Comments for the Author):

The authors develop a transformation system and laccase-based reporter system for a brown rot fungus *G. trabeum*. The authors have a track record of developing resources for the wood-rotting fungal community and are experts with brown rots and this particular brown rot. The work seems technically sound with sufficiently detailed methods, but there is one major issue related to the previous literature that I have concerns about.

The following study does not seem to be cited by the authors? This must be rectified by the authors. This study describes a

genetic transformation system for the same fungus. It undermines some of the novelty. I understand *Microbiology Spectrum* is not novelty-focused, but the relevant literature must be considered. On Line 16 the authors state, "However, there has been no tractable brown-rot strain in which to..." but the below paper seems to call this into question.

Also, this paper seems to undermine another claim of this study whereby they "express an endogenous (*G. trabeum*) laccase candidate gene (*Gtlcc3*) under...". The reporter system claimed in the submitted manuscript is laccase-based and works partly because on line 191 the authors state, "laccase ... and its absence of enzyme activities in brown rot fungi....".

Apologies if there is a misunderstanding about species names etc, which can frequently be revised.

Arimoto M, Yamagishi K, Wang J, Tanaka K, Miyoshi T, Kamei I, Kondo R, Mori T, Kawagishi H, Hirai H. Molecular breeding of lignin-degrading brown-rot fungus *Gloeophyllum trabeum* by homologous expression of laccase gene. *AMB Express*. 2015 Dec;5(1):81. doi: 10.1186/s13568-015-0173-9. Epub 2015 Dec 22. PMID: 26695948; PMCID: PMC4688280.

Line 1: the title must include the name of the fungus, and also adding that it is a laccase-based reporter system would be more informative to the reader.

Line 86: It should be stated clearly in M&Ms that the strain is a monokaryon.

Line 152: It states that mitotic stability was verified by five successive transfers on hygromycin. Did this include a type of ON/OFF selection sub-culturing, i.e., subculture onto hyg, then sub-culture onto the plate without hyg, and then sub-culture to another plate with hyg. This can help verify if the transformant is pure and also that the resistance gene will not be "lost".

Line 154: Did the authors also perform a Southern blot to confirm integration into the basidiomycete genome? I can see they did for the laccase constructs later.

Line 216: The viability of the protoplasts is described. But did the authors also check the authenticity of the protoplasts e.g., if the protoplasts were diluted in water, and plated on medium without an osmotic stabilizer, there should not be any colonies if all of what were considered protoplasts were real/authentic protoplasts. This is more of a problem when germinating spores are used for protoplasting as a non-germinated spore can be mistaken for a protoplast.

Line 261: "was sent" is not the best expression.

Line 424: The authors should discuss more how the laccase would compare as a reporter gene to luciferase, GFP, etc. Especially considering that background laccase activity is partly assumed not to be a problem because the laccase-like genes in *G. trabeum* are not expressed. I think GFP can also be similarly quick to laccase if one uses an epi-fluorescent binocular type microscope for screening plates. I think it is worth noting that the laccases may be developmentally regulated, such as during fructification, but this will not be a problem with monokaryon.

Fig. 4, part (E), here the relative protoplast regeneration % seem quite high but in line 216 it states that the protoplast regeneration rate was 1-2%. If the values are relative to one treatment, instead I think it is better to show the actual regeneration percentages.

I think depositing the vectors described in this work in the Addgene database can be very useful for other researchers, but it is of course not essential for publication.

Did the authors check how many nuclei are present in the protoplasts, such as by Hoechst stain or SYBR green? It is not directly relevant to this study but can be useful for future research on gene deletions. e.g., if the protoplasts are multinucleate, it can be difficult to purify transformants. The type of transformation work described by the authors does not confirm that their transformants are genetically pure, but it is probably unlikely that wild type cells are maintained if the antibiotic concentration is high enough.

Sorry if I have missed this, was a blender ever used in the mycelia preparation? Blenders such as Waring blenders are commonly used with wood-rotting fungi.

I'm curious, is this work also the subject of any patent applications, as transformations of brown rots also has commercial applications?

Reviewer #2 (Comments for the Author):

The authors report the construction of gene manipulation system in the model brown-rot fungus *Gloeophyllum trabeum*. The easy-to-detect laccase was successfully used as a reported gene. The research was well designed and clearly presented in the

manuscript. The system is expected to be used for studying the mechanisms of lignocellulose degradation as well as other biological processes in this important fungus. The following issues should be addressed.

1. The preparation of protoplasts is key for DNA transformation. Did the authors try the use of only one of the two lysing enzymes, or lysing enzymes of other sources? Such details/notes in transformation would be very helpful for other researchers in this field.
2. As I know the genetic manipulation tools in some basidiomycetous fungi are still quite limited, despite the report of successful expression of commonly used reporter genes. I suggest the authors provide more discussions on such difficulties or challenges in future development of gene targeting and gene editing methods in *Gloeophyllum trabeum* (and maybe basidiomycetous fungi).
3. How to understand the different sizes of the colonies of transformants in Fig. 2B?
4. Line 99. I would not agree with that the ABTS-medium was used for the screening of transformants. It seems this plate was only used for phenotypic analysis of transformants screened out using hygromycin B.
5. Line 110. The accession number of pMD-1 or the sequence of HygR on this plasmid should be provided.
6. Line 138. Novoenzyme should be Novozymes.
7. Line 143. Better use "production" to replace "productivity".
8. Line 185. subscript T is usually used in CT.
9. Line 268. Why mention or highlight the nitrogen content of HMM?
10. Figs. 1 and 2. gdp should be gpd. Kb should be kb (also correct in the main text).
11. Fig. 2D. Images from different experiments should not be spliced together. A space can be left between them.

Staff Comments:

Preparing Revision Guidelines

Please return the manuscript within 60 days; if you cannot complete the modification within this time period, please contact me. If you do not wish to modify the manuscript and prefer to submit it to another journal, please notify me of your decision immediately so that the manuscript may be formally withdrawn from consideration by Microbiology Spectrum.

First, we appreciate the editor's and reviewers' constructive feedback and contributions to our manuscript. We believe this review process will have helped improve this manuscript and advance our next-step brown rot phenotyping research in this emerging model species. According to reviewers' comments, we have revised the manuscript and provide the following responses, point-by-point:

Responses to Reviewer #1:

Comments: *The authors develop a transformation system and laccase-based reporter system for a brown rot fungus *G. trabeum*. The authors have a track record of developing resources for the wood-rotting fungal community and are experts with brown rots and this particular brown rot. The work seems technically sound with sufficiently detailed methods, but there is one major issue related to the previous literature that I have concerns about.*

The following study does not seem to be cited by the authors? This must be rectified by the authors. This study describes a genetic transformation system for the same fungus. It undermines some of the novelty. I understand Microbiology Spectrum is not novelty-focused, but the relevant literature must be considered. On Line 16 the authors state, "However, there has been no tractable brown-rot strain in which to..." but the below paper seems to call this into question.

*Also, this paper seems to undermine another claim of this study whereby they "express an endogenous (*G. trabeum*) laccase candidate gene (*Gtlcc3*) under...". The reporter system claimed in the submitted manuscript is laccase-based and works partly because on line 191 the authors state, "laccase ... and its absence of enzyme activities in brown rot fungi....".*

*Apologies if there is a misunderstanding about species names etc, which can frequently be revised. Arimoto M, Yamagishi K, Wang J, Tanaka K, Miyoshi T, Kamei I, Kondo R, Mori T, Kawagishi H, Hirai H. Molecular breeding of lignin-degrading brown-rot fungus *Gloeophyllum trabeum* by homologous expression of laccase gene. *AMB Express*. 2015 Dec;5(1):81. doi: 10.1186/s13568-015-0173-9. Epub 2015 Dec 22. PMID: 26695948; PMCID: PMC4688280.*

Response: This is a very helpful citation addition, and to avoid any overstatements, we have toned down our language in some lines, with an example, as follows: "*However, there has been a significant lack of genetic tools in brown-rot species by which to...*" in revised lines 16-17, 28, and throughout the manuscript. In terms of this referred paper, we feel it is not reducing the novelties of the gene reporting system we developed, but instead it does a very good job supporting working within this species. It is a nice citation addition. In our paper, we primarily focused on the development of a laccase-based gene reporting system and its applications for optimizing transformation methods. We feel this is prominently distinct from the referred paper where the "molecular breeding" was the main goal. With this context, we reported a large number of optimized details on how to establish a more robust genetic transformation system, which we believe will further support others' research in this field. Given its clear support to our research, we have cited this suggested paper revised lines 317-.

The use of laccase as the reporter gene has been validated and stated from several aspects throughout the manuscript. We have summarized these here to highlight: 1) the absence of native laccase genes' expression in wild-type *G. trabeum* ATCC 11539 strain by qPCR and RNA-seq (Fig. 2E; Zhang et al., 2019, mBio), 2) the absence of WT laccase activities under variable C and N conditions (Fig. 4), and 3) we have been using a heterologous laccase gene obtained from a white-rot fungus *Trametes versicolor* as the reporter gene. In addition, this absence of WT laccase expression is also true in the reviewer referred citation, where the native laccase gene *Gtlcc3* has

to be forced to express by the function of a strong, constitutive *gpd* promoter in *G. trabeum* KU41. This again convincingly demonstrates that the laccase is a reliable reporter in *G. trabeum* strains.

Comments: *Line 1: the title must include the name of the fungus, and also adding that it is a laccase-based reporter system would be more informative to the reader.*

Response: We have changed the title to “A Laccase Gene Reporting System that Enables Genetic Manipulations in a Brown-Rot Wood Decomposer Fungus *Gloeophyllum trabeum*.” (in the revised line 1)

Comments: *Line 86: It should be stated clearly in M&Ms that the strain is a monokaryon.*

Response: We have added this information in the revised manuscript. (lines 89-)

Comments: *Line 152: It states that mitotic stability was verified by five successive transfers on hygromycin. Did this include a type of ON/OFF selection sub-culturing, i.e., subculture onto hyg, then sub-culture onto the plate without hyg, and then sub-culture to another plate with hyg. This can help verify if the transformant is pure and also that the resistance gene will not be "lost".*

Response: Yes, we did use the selective and non-selective plates to confirm the mitotic stabilities of transformants. Occasionally, a few can form irregular colonies and need to be abandoned for downstream analyses. We also stored the transformants on the non-selective slants in freezer (-20°C) and then revived them on hygromycin B plates. These tests can confirm the stability of the resistance gene. The relevant methods have been clarified and listed in revised lines 154-.

Comments: *Line 154: Did the authors also perform a Southern blot to confirm integration into the basidiomycete genome? I can see they did for the laccase constructs later.*

Response: No, we did not check the copy numbers of the inserted hygromycin-resistant gene. Instead, we mostly focused on the validation of the presence/absence of hygromycin-resistant functions in transformants. However, we performed Southern blotting for the laccase gene to study the correlation between copy number of insertion and its influence on expression level.

Comments: *Line 216: The viability of the protoplasts is described. But did the authors also check the authenticity of the protoplasts e.g., if the protoplasts were diluted in water, and plated on medium without an osmotic stabilizer, there should not be any colonies if all of what were considered protoplasts were real/authentic protoplasts. This is more of a problem when germinating spores are used for protoplasting as a non-germinated spore can be mistaken for a protoplast.*

Response: This is not a major problem in our method, as we used young, active hyphae for protoplasting. Protoplast cells will immediately break after mixed with ddH₂O during microscopic monitoring, and only a few colonies per 10⁷ protoplasts can be regenerated on plates without sucrose as the osmotic stabilizer, which are likely from the undigested mycelia.

Comments: *Line 261: "was sent" is not the best expression.*

Response: This has been changed to “... was delivered into...” (line 241)

Comments: *Line 424: The authors should discuss more how the laccase would compare as a reporter gene to luciferase, GFP, etc. Especially considering that background laccase activity is partly assumed not to be a problem because the laccase-like genes in *G. trabeum* are not expressed. I think GFP can also be similarly quick to laccase if one uses an epi-fluorescent binocular type microscope for screening plates. I think it is worth noting that the laccases may be developmentally regulated, such as during fructification, but this will not be a problem with monokaryon.*

Response: Yes, these comparisons of advantages/disadvantages of different reporter genes have been discussed in lines 333-346. In this discussion and by other results obtained from this work, we justified the use of laccase as our first-generation reporter in *G. trabeum*. We agree that GFP could potentially serve as a reporter, but it might need codon optimization for expression in cross-species. Recently, we codon-optimized a GFP gene and fused it to Cas9 for cellular localization, which has been gaining some successes. However, GFP signal visualization needs expensive fluorescent microscopy and the process takes longer than using laccase reporter in our current procedure.

Comments: *Fig. 4, part (E), here the relative protoplast regeneration % seem quite high but in line 216 it states that the protoplast regeneration rate was 1-2%. If the values are relative to one treatment, instead I think it is better to show the actual regeneration percentages.*

Response: The regeneration rates in Fig. 4E are normalized values by setting the fresh protoplasts as 100%. With this, we meant to emphasize the “changes” caused by different preservation treatments. Considering reviewer’s comment, the actual regeneration rates ($1.01 \pm 0.47\%$) have been added to figure E captions for the fresh control.

Comments: *I think depositing the vectors described in this work in the Addgene database can be very useful for other researchers, but it is of course not essential for publication.*

Response: This is a great idea, and we can do this by affiliating the publication record.

Comments: *Did the authors check how many nuclei are present in the protoplasts, such as by Hoechst stain or SYBR green? It is not directly relevant to this study but can be useful for future research on gene deletions. e.g., if the protoplasts are multinucleate, it can be difficult to purify transformants. The type of transformation work described by the authors does not confirm that their transformants are genetically pure, but it is probably unlikely that wild type cells are maintained if the antibiotic concentration is high enough.*

Response: This is our concern too. Although it is beyond the current scope, nucleotype would influence gene editing processes, as stated by the reviewer. However, visualizing nuclei in protoplast has been very challenging since the UV wavelength used for stimulating these fluorescent dyes will immediately break down the protoplast cell. We hope to overcome this challenge by continually optimizing the microscopic methods.

For the concern that the transformant could be contaminated by the WT cells, we also feel this is an unlikely event since we have used ON/OFF screenings to test the mitotic stability of mutants, as our responses to one of the above comments.

Comments: *Sorry if I have missed this, was a blender ever used in the mycelia preparation? Blenders such as Waring blenders are commonly used with wood-rotting fungi.*

Response: We were not using blenders in our procedures, as we found that without blending would make the procedure less complicated and easier to handle.

Comments: *I'm curious, is this work also the subject of any patent applications, as transformations of brown rots also has commercial applications?*

Response: Thank you for this kind reminder, and we will consult with our Office for Technology Commercialization for the patent application.

Responses to Reviewer #2:

Comments: *The authors report the construction of gene manipulation system in the model brown-rot fungus *Gloeophyllum trabeum*. The easy-to-detect laccase was successfully used as a*

reported gene. The research was well designed and clearly presented in the manuscript. The system is expected to be used for studying the mechanisms of lignocellulose degradation as well as other biological processes in this important fungus. The following issues should be addressed.

Response: We appreciate reviewer's encouraging comments, and thank you!

Comments: 1. The preparation of protoplasts is key for DNA transformation. Did the authors try the use of only one of the two lysing enzymes, or lysing enzymes of other sources? Such details/notes in transformation would be very helpful for other researchers in this field.

Response: Yes, we tried the use of only one of the two mentioned lysing enzymes, but it turned out that the combination of Yatalase/Sigma L1412 and VinoTaste Pro performed better than using a sole enzyme component. We also tested different enzyme concentrations, and these optimized conditions were listed in the methods. (line 139-, line 214-)

Comments: 2. As I know the genetic manipulation tools in some basidiomycetous fungi are still quite limited, despite the report of successful expression of commonly used reporter genes. I suggest the authors provide more discussions on such difficulties or challenges in future development of gene targeting and gene editing methods in *Gloeophyllum trabeum* (and maybe basidiomycetous fungi).

Response: We agree with the reviewer that there are still many challenges for genetic manipulation in *Basidiomycete* fungi, which have been discussed in our manuscript in lines 322-. These challenges are particularly evident in the development of genome-editing tools in *G. trabeum* and other *Basidiomycete* fungi. We have cited a recent review paper and our ongoing efforts to notify this. (lines 379-)

Song R, Zhai Q, Sun L, Huang E, Zhang Y, Zhu Y, Guo Q, Tian Y. 2019. CRISPR / Cas9 genome editing technology in filamentous fungi: progress and perspective. *Appl Microbiol Biotechnol* 103:6919–6932.

Comments: 3. How to understand the different sizes of the colonies of transformants in Fig. 2B?

Response: The variables of colony size or morphological phenotype of transformants are likely caused by the random insertions of the *HygR* and *TvLac3* genes into the genomes, as indicated by the Southern blotting. We have discussed these effects of random insertions in lines 373-.

Comments: 4. Line 99. I would not agree with that the ABTS-medium was used for the screening of transformants. It seems this plate was only used for phenotypic analysis of transformants screened out using hygromycin B.

Response: We have changed this description, as in lines 100-.

Comments: 5. Line 110. The accession number of *pMD-1* or the sequence of *HygR* on this plasmid should be provided.

Response: As mentioned by another reviewer, we will upload our plasmids used in this work to addgene, making them publicly available.

Comments: 6. Line 138. Novoenzyme should be Novozymes.

Response: This has been corrected.

Comments: 7. Line 143. Better use "production" to replace "productivity".

Response: This has been corrected.

Comments: 8. Line 185. subscript *T* is usually used in CT.

Response: This has been changed through the manuscript.

Comments: 9. Line 268. Why mention or highlight the nitrogen content of HMM?

Response: These are the commonly used concentrations (high/low) for testing N effects in wood decay fungi. We feel this is not necessary to be highlighted here, so we have deleted this information in the revised line 266.

Comments: 10. *Figs. 1 and 2. gdp should be gpd. Kb should be kb (also correct in the main text).*

Response: This has been corrected in the revised figures and the main text.

Comments: 11. *Fig. 2D. Images from different experiments should not be spliced together. A space can be left between them.*

Response: This has been changed according to the comment.

December 22, 2022

Dr. Jiwei Zhang
University of Minnesota Twin Cities
Bioproducts and Biosystems Engineering
2004 folwell ave
Saint Paul, MN 55108

Re: Spectrum04246-22R1 (A Laccase Gene Reporting System that Enables Genetic Manipulations in a Brown-Rot Wood Decomposer Fungus *Gloeophyllum trabeum*)

Dear Dr. Jiwei Zhang:

Your manuscript has been accepted, and I am forwarding it to the ASM Journals Department for publication. You will be notified when your proofs are ready to be viewed.

Sincerely,

Chengshu Wang
Editor, Microbiology Spectrum
